# Health worker perspectives on access to antenatal care in rural plains Nepal during the COVID-19 pandemic

Bibhu Thapaliya[1], Samata Kumari Yadav[1], Sanju Bhattarai[1], Santosh Giri[1], Suprich Sapkota[1], Abriti Arjyal[1], Helen Harris-Fry[2], Naomi Saville[3], Sara Hillman[4], Sushil Baral[1], Joanna Morrison[3]*

1 HERD International, Lalitpur, Nepal, 2 London School of Hygiene & Tropical Medicine, Keppel St, London, United Kingdom, 3 UCL Institute for Global Health 30 Guilford Street, London, United Kingdom, 4 UCL Institute for Women's Health Room 237c Medical School Building, London, United Kingdom

* Joanna.morrison@ucl.ac.uk

## Abstract

The COVID-19 pandemic affected access to antenatal care in low and middle-income countries where anaemia in pregnancy is prevalent. We analyse how health workers provided antenatal care and the factors affecting access to antenatal care during the COVID-19 pandemic in Kapilvastu district in the western plains of Nepal. We used qualitative and quantitative methodologies, conducting eight semi-structured interviews with health workers who provided antenatal care during the pandemic, and a questionnaire containing open and closed questions with 52 female community health volunteers. Antenatal care was severely disrupted during the pandemic. Health workers had to find ways to provide care with insufficient personal protective equipment and guidance whilst facing extreme levels of stigmatisation which prevented them from providing outreach services. Pregnant women were fearful or unable to visit health institutions during the pandemic because of COVID-19 control measures. Pre-pandemic and during the pandemic health workers tried to contact pregnant and postpartum women and families over the phone, but this was challenging because of limited access to phones, and required pregnant women to make at least one antenatal care visit to give their phone number. The pandemic prevented new pregnancies from being registered, and therefore the possibilities to provide services over the phone for these pregnancies were limited. To reach the most marginalised during a pandemic or other health emergency, health volunteers and households need to exchange phone numbers, enabling proactive monitoring and care-seeking. Strengthening procurement and coordination between the municipal, provincial, and federal levels of government is needed to ensure adequacy of antenatal supplies, such as iron folic acid tablets, in health emergencies. Community engagement is important to ensure women and families are aware of the need to access antenatal care and iron folic acid, and to address stigmatisation of health workers.

**Data Availability Statement:** Participants consented to participate in the study with guarantee of anonymity, but due to the limited and known number of health facilities in the district,

anonymity cannot be guaranteed if we make the data set publicly available. Therefore, data cannot be shared publicly for ethical reasons. Ethical restrictions are applied by the Nepal Health Research Council (ethicalreviewb2@gmail.com), the UCL Ethics Committee (ethics@ucl.ac.uk), and the London School of Hygiene & Tropical Medicine ethics committee (Ethics@lshtm.ac.uk).

**Funding:** The study was funded by the UK Medical Research Council (MRC/RO20485/1) (Sara Hillman was the Principal Investigator for this grant, but all authors were funded through this grant except HHF) and HHF was funded through a Sir Henry Wellcome Fellowship (210894/Z/18/Z).

**Competing interests:** The authors have declared that no competing interests exist.

# Introduction

Globally, maternal, and foetal health outcomes worsened during the COVID-19 pandemic [1]. A systematic review found that general health care utilisation decreased by one-third [2], and a modelling study predicted that antenatal care (ANC) uptake could decrease by at least 18% because of disruptions to care and difficulties accessing services [3]. ANC is an essential component of comprehensive maternity care. Evidence from low- and middle-income countries (LMICs) shows that ANC is associated with reduced chance of newborn and infant mortality, low-birth weight, stunting and underweight [4]. ANC can also prevent anaemia in pregnancy which is one of the main causes of maternal deaths and adverse pregnancy outcomes in LMICs [5, 6]. Iron folic acid (IFA) distribution and nutrition counselling are key components of ANC. Osendarp et al. estimated that from 2020 to 2022 the pandemic could cause an additional 2.1 million maternal anaemia cases because of decreased access to health services [7].

The WHO issued guidance on the maintenance of essential maternal health services during the COVID-19 pandemic and advised that these should be adapted to local contexts. Adaptions should consider local disease burden, social context, the COVID-19 transmission scenario, and local capacity to provide services [8]. Despite this, it was often difficult for health systems in LMICs to respond because of a lack of health system resilience and the unwillingness of women and families to use services [9, 10]. It is important to understand how health workers coped during the COVID-19 pandemic and explore the factors affecting ANC provision to develop more resilient health systems. We analysed health workers' experiences of providing ANC during the COVID-19 pandemic in rural Nepal, with a particular focus on the prevention of anaemia in pregnancy and discuss the implications of this for future emergency response planning.

## ANC and anaemia in pregnancy in Nepal

ANC is provided at Health Posts, Primary Health Centres and at monthly outreach clinics [11]. According to government protocol, pregnant women should have four ANC check-ups. At these check-ups they should receive iron-folic acid (IFA), calcium, and de-worming tablets, have their weight measured, their blood pressure checked, have a Haemoglobin test for anaemia, tetanus and diphtheria vaccination and receive counselling on healthy eating [12]. ANC is usually provided by paramedics (Auxiliary Nurse Midwives (ANMs)) or nurses. ANC is also available at outreach clinics attended by Female Community Health Volunteers (FCHVs). FCHVs are locally resident women with at least 18 days of training who are responsible for community health promotion. They run monthly mother's groups, identify and refer pregnant women to health institutions, and re-supply IFA [13]. Nationally, 84% of pregnant women are reported to have attended at least one check-up but there are large urban-rural differences in access, number, and timing of antenatal visits [14]. Access is related to education, socioeconomic status [15], caste and ethnicity [16, 17]. Qualitative studies show that access is also influenced by family relationships and gender norms which limit agency, access to resources, and restrict movement outside the home [18, 19]. Health workers and FCHVs are responsible for nutrition counselling during ANC, but there are indications that the quality of this counselling may be sub-optimal [20, 21]. Despite the following global guidance to address anaemia in pregnancy, anaemia is an intractable problem in Nepal. The National Demographic Health Survey (NDHS) 2016 reported anaemia in 46% of pregnant women which was only a 2% decrease from 2011 [22].

## ANC during the pandemic in Nepal

The first COVID-19 case was detected on 23 January 2020 in Nepal and there was a nationwide lockdown from 23rd March to 21st July 2020 which restricted travel, closed businesses,

and prohibited gatherings [23]. From 23rd January 2020 to 22nd January 2021, there were 1986 COVID related deaths [24]. Another surge in cases occurred from May 2021 [25] resulting in 6,699 COVID related deaths between mid-March and late July 2021 [26]. During this time, lockdowns were managed locally (as opposed to nationally) as part of a smart lockdown strategy to contain transmission and to minimise effects on service provision, business, and movement where possible [27]. Health services were overwhelmed, and oxygen was in short supply [28–30].

Nepal's Ministry of Health and Population emergency preparedness plan was published on 7th May 2020. Technical guidelines, standard operating procedures, tools, and training materials to manage COVID-19 were made for the federal, provincial and municipal governments by August 2020 [23]. The preparedness plan explicitly mentioned the need to ensure adequate Personal Protective Equipment (PPE) for health workers and maintain basic and emergency healthcare including ANC services during lockdowns at federal, provincial and municipal levels [31]. However, plans did not specify how health facilities should access PPE. Instead, resources were to be used on quarantine management, community engagement and risk communication, case investigation and contact tracing, surveillance, screening at points of entry, community-level screening and testing, and emergency response teams [31]. The plan also warned of the legal implications of any discrimination, violence, or harassment of health workers [31]. On 21 May 2020, interim guidance for the provision of reproductive, maternal, newborn and child health (RMNH) services was approved by Family Health Division [32]. This guidance stated that ANC services should be provided while maintaining a physical and social distance [32]. It acknowledged problems of procurement, but also emphasised the importance of providing adequate PPE, sanitiser, and using a single entrance for patients and visitors. Health workers were instructed to collect phone numbers of pregnant women through the FCHV and call them to conduct 'tele ANC'. During this call, health workers should ask questions about general well-being, advise pregnant women about how ANC was being managed at the health facility, and tell them how to prevent COVID-19. Pregnant women were advised to come to health facility for ANC by FCHVs and health workers (Ministry of Health and Population, 2020). There was no guidance about how pregnant women could receive ANC when they could not reach the health facility.

Whilst we know that access to maternal health care declined during the pandemic, we lack understanding of how health workers responded to the pandemic, how they provided services, and how demand and supply side issues affected ANC. Our study uses qualitative and quantitative data from health workers in Kapilvastu District to describe this experience and discuss implications for future planning.

## Materials and methods

### Study setting

We collected data in Kapilvastu district, Lumbini Province, in the western plains of Nepal, bordering Uttar Pradesh state of India. The district population is ~570,000 and literacy rates of women and men are 45% and 65% respectively [33]. Most of the population are Hindu Madhesi's many of whom speak Awadhi language. Anaemia prevalence is high (45%) and the number of women taking IFA in pregnancy is sub-optimal: in 2017 43% took at least the recommended dose of 180 IFA tablets, 33% took 60–179 tablets, and 24% took <60 tablets [22].

COVID-19 cases reached their peak in Lumbini Province in October 2020 and at that time, Kapilvastu District had the highest number of COVID-19 cases in the province [34]. The district extended its lockdown for around a month after most districts had lifted restrictions [35].

Many believed that the high number of migrant workers returning from India to Nepal through (and to) Kapilvastu was partly to blame for the high numbers of COVID-19 cases [36], and the district was criticised for poor management of isolation and quarantine centers during the first phase of the pandemic (TKP 2020a). Isolation and quarantine management was problematic in many part of Nepal [34, 37]. The District Security Committee closed all public service offices, businesses, transport, and banks from August 16th, 2020.

## The effect of COVID-19 on ANC uptake in Kapilvastu

We analysed four years' health system data (DHIS-2) from June 2017 –June 2021 to describe how ANC use fluctuated before, during and after the pandemic (Fig 1). We were unable to obtain the latest figures for 2022 as the data was not public at the time of writing. We obtained DHIS-2 data in Microsoft Excel from the District Health Office and annual reports from Department of Health Services. We cleaned the data and prepared the trend visualisations in Excel. In Kapilvastu 63.1% of pregnant women made four ANC visits pre-pandemic, which was higher than the national figure of 56.3%. During the pandemic the percentage of pregnant women who had four ANC visits nationally and in Kapilvastu dropped to 52.3% and 52.3%, respectively, and the percentage of pregnant women having four ANC visits had still not returned to pre-pandemic levels in 2021, remaining low at 54.9%.

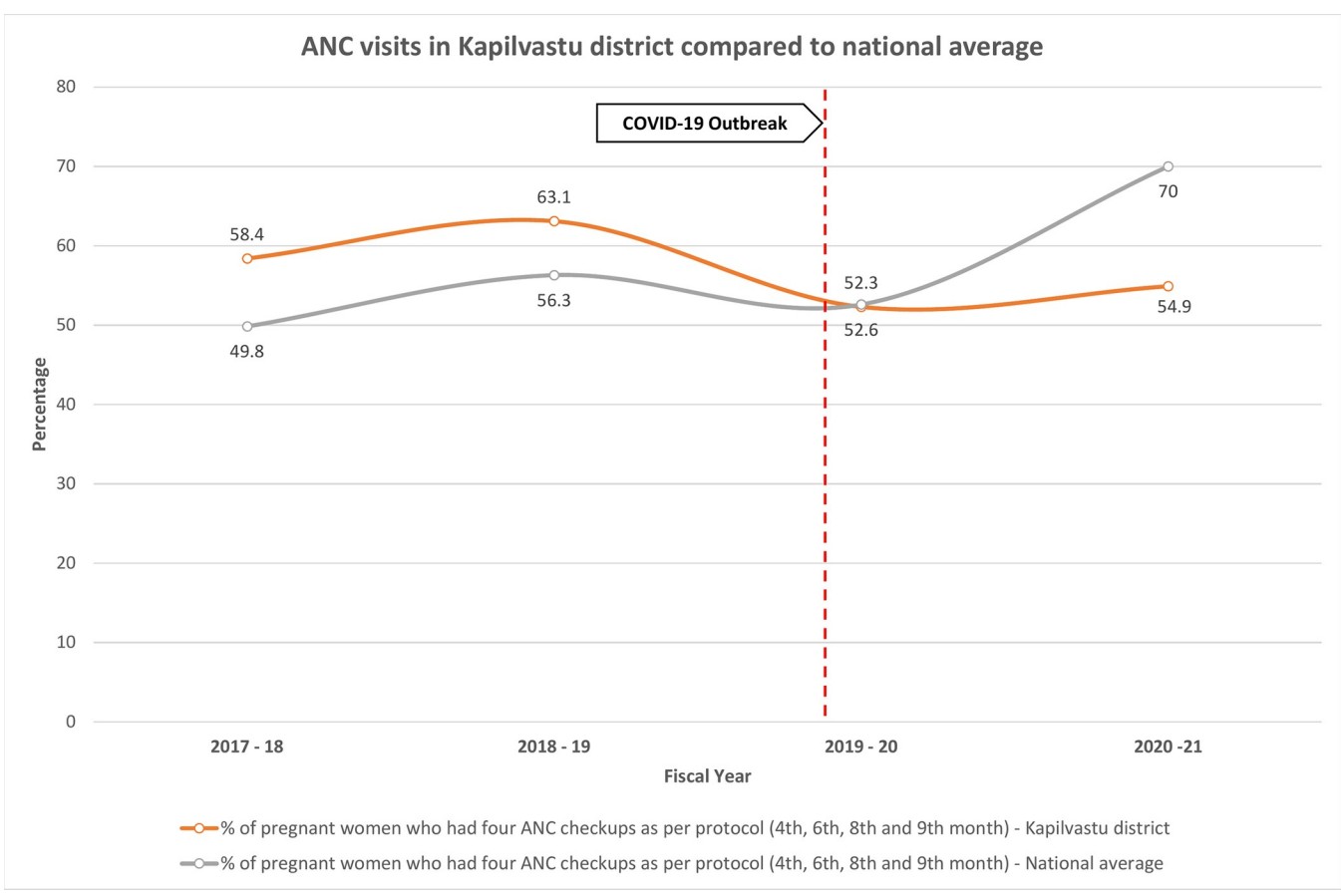

**Fig 1. ANC visits in Kapilvastu district compared to national average.**

## Qualitative sampling

We recruited health workers from eight conveniently sampled health posts that were being visited for another study [38]. We purposively sampled health workers who had provided ANC during the pandemic. Health workers had worked between one and 22 years in their health institution. No one refused to participate in the research.

## Qualitative data collection

We conducted semi-structured interviews (SSIs) using topic guides with three Auxiliary Nurse Midwives (ANM), one senior ANM, one Auxiliary Health worker (AHW), two senior AHW, and one Public Health Inspector (in-charge) about how services were provided during the COVID-19 pandemic (March-July 2020) (Table 1). Interviews were face-to-face and took around one hour. Data were collected in Nepali language and recorded. Data were collected between February and May 2022 by three trained qualitative researchers (one male and two female) who were from the local area.

## Qualitative data management and analysis

Interviews were transcribed verbatim into Nepali and then translated into English. Three senior female researchers (BT, SY, JM) read the translated interviews and discussed and identified themes in the data. BT and SY each coded half of the data in Microsoft Excel, and JM coded all the data in Nvivo. We compared our coding and discussed macro level themes. JM and BT then wrote a narrative description of these themes.

## Quantitative sampling, data collection, management, and analysis

In addition to the qualitative data, we surveyed half of the FCHVs working under the 8 health institutions used for the health worker interviews. We asked closed questions about how many women's groups and outreach clinics they had conducted during the pandemic, and how many women had received IFA from them between January 2020 and December 2021. Interviews were face-to-face and took around one hour. Data were collected on paper forms, entered in Microsoft Excel and tabulated. We also asked three open questions about the factors affecting their ability to deliver services during the pandemic. These data were also collected on paper forms, and directly entered in Excel and translated to English. JM, BT and SY read and discussed these data, and then BT and SY coded data in Microsoft Excel using the macro level themes previously identified. BT, SY and JM compared data from health workers and FCHVs, and JM added to the narrative description of health worker data and discussed this descriptive report with collaborators.

**Table 1. Data collection.**

| Participant | Gender | N |
|---|---|---|
| **Qualitative sample** | | |
| ANM | Female | 3 |
| Senior ANM | Female | 1 |
| AHW | Male | 1 |
| Senior AHW | Male | 2 |
| Public Health inspector (in-charge) | Male | 1 |
| **Quantitative sample** | | |
| FCHV | Female | 52 |
| **Total** | | **60** |

## Ethical considerations

Participants gave written or verbal informed consent to participate, depending on their literacy, and the study received ethical approval from the Nepal Health Research Council (353/2019) the UCL Ethics Committee (14301/001) and London School of Hygiene & Tropical Medicine ethics committee (16528).

## Results

### Supply side factors affecting ANC during the pandemic

**Fear of infection.** At the beginning of the pandemic, most community health institutions were closed. Most health workers said that they stopped giving services because they were afraid of infection: "We were forced to stop ANC services because we did not have PPE. We could not expose ourselves and pregnant women to the virus" (Health worker 06). They were also worried about infecting their own families. Health workers also said that many people were unconcerned about COVID-19, which made it hard for them to provide services: "We were very scared in first phase of COVID. (but) people here were not scared then. They criticised us when we asked them to wear a mask. They said: 'why should we wear a mask? It's just a rumour'" (Health worker 06).

**Doing what was possible.** Health institutions resumed services after some time–this varied by health institution—but with reduced capacity. When health workers received masks and PPE these were often insufficient and local help was slow to be received: "No organisation came forward to help" (Health worker 01). In one health institution, they called the police to help them manage social distancing and queuing for services. Health workers said there was a lack of guidance about how to provide services, and they did what was possible within the constraints of their environment: "Everyone panicked at that time. No plans were prepared; we did what was directed from the upper level. We would have managed pregnant women separately if we could have. . .If their entry and exit would have been different, they would not have had contact with other visitors of the health post minimising the risk of the virus spreading" (Health worker 10).

**Quality affected by lack of PPE.** All but two health workers felt that the quality of services suffered in the early phase of the pandemic: "We couldn't give quality services earlier. Pregnant women didn't receive quality services. We were not able to meet their expectations" (Health worker 06). Health workers and FCHVs said that pregnant women were dissatisfied with the poor-quality ANC: "For those who went for ANC check-up, it was a frustrating experience. They would get ANC cards but not get any physical examination. They felt ANC cards were just for the government's purpose of recording their health status, and not actually for their service" (FCHV). Some FCHVs said it was challenging to refer pregnant women, as health institutions were either not providing services, or not providing them comprehensively. The cost of transport was also high. One FCHV said: "It was difficult to take women to the hospital. The health workers would not do a proper (ANC) check-up. They were afraid to even touch the patients. And even though the services were not delivered properly, they still signed in the ANC cards and told women to go home. During lockdown, pregnant women did not receive normal services" (FCHV).

Any services that required physical contact were not delivered during the early phases of the pandemic, and health workers specifically discussed not being able to give tetanus injections, take blood pressure or weight measurements. Health workers provided IFA to pregnant women through the window of the health institution, and some made a pulley system to dispense medicines through the window: "We dispensed medicines by tying them to strings to avoid touching people" (Health worker 11). One health worker described the way that they

gave services: "We couldn't give service directly. Pregnant women brought their ANC cards, and we took them through the window. We couldn't even check their blood pressure and weight. We checked them by looking at their physical appearance, like checking for swelling. We asked if they are having pain and giving medicine. We counselled them about nutrition, hygiene and how to prevent COVID-19" (Health worker 05). When health workers had access to COVID-19 vaccines and PPE they felt more able to provide services: "Pregnant women and health workers were scared. Now we are providing health services, as we have adequate availability of masks and gloves" (Health worker 06).

**Shortage of supplies.**   Only one health institution out of 8 had a shortage of IFA, all others said that they had enough IFA and de-worming tablets to provide pregnant women: "Due to a shortage of iron tablets, we could only provide them for a month. We continuously provided IFA but due to a lack of procurement of medicines, there were difficulties" (Health worker 08). Other services such as Prevention of Mother to Child Transmission of HIV (PMTCT) test kits and tetanus injections were in short supply in many health institutions and therefore those services were not provided. Survey data show that FCHVs were often unable to resupply IFA to pregnant women. Strategies to enable access to IFA included: rationing IFA to split among more women; health workers delivering IFA to FCHVs in the community; or FCHVs accompanying pregnant women to the health facility to receive IFA.

**Re-deployment of human resources.**   Some managerial level health workers (Senior Auxiliary Health Worker and Health Inspector) said that there was a shortage of health workers to provide services as they were deployed to work in quarantine facilities: "The environment was different at that time. Everyone was scared. We had a shortage of staff. We had to look after quarantine as well as provide services in the health post" (Health worker 05). Some felt that this affected the quality of service they were able to provide: "Initially, we had only a paramedic (ANM) in the health institution. The service provided by paramedics can't be good as that provided by those with nursing qualifications. It's the same as care given by a father will not be good enough as care given by a mother" (Health worker 07). Redeployment did not affect all health institutions and one senior ANM said that this wasn't an issue for their health institution.

**Limited community outreach.**   We asked 52 FCHVs if they distributed IFA between January and December 2021. 60% did not re-supply IFA despite this being part of their normal work pre-pandemic. A few FCHVs were in contact with pregnant women over the phone: "I made phone calls and talked to pregnant women. I suggested them to go to hospital in case there was any problem. I even took some of them to the hospital by myself" (FCHV). Many FCHVs provided home-based counselling of pregnant women wearing masks and maintaining social distancing. They found it difficult to travel due to increased costs and lockdowns. They usually walked to health facilities and around communities or travelled after taking permission from the local police.

Mother's group meetings were also affected by COVID-19. From January to December 2021, 34 out of 52 FCHVs held mothers group meetings. Seven FCHVs held 12 meetings and three FCHVs conducted 10 meetings. Those who conducted meetings said they gathered small groups of women, used masks, and ensured social distancing was followed: "There were just 4 or 5 of us in the meetings. I bought and distributed masks and maintained a distance of 2 meters" (FCHV).

## Demand side factors affecting ANC during the pandemic

**Transport and lockdown.**   Public transport was limited which made it difficult for pregnant women to travel to health institutions: "Very few pregnant women came to the health

post at that time. Most had run out of IFA tablets but still couldn't visit the health post" (FCHV). A few health workers were concerned about the low vaccination coverage and attendance of pregnant women and went to villages to motivate people to come to the health institution. Community members were often hostile: "When we visited their homes as they didn't visit for the check-up according to our records, they told us to go away and even chased us in the fields. Although there were challenges, we didn't give up" (Health worker 11).

**Stigmatisation of health workers.** Pregnant women were fearful of getting COVID-19 from health workers. Health workers experienced animosity when they went to villages: "People were scared that we would transmit the virus to them. A local man asked why I was in his community implying that I shouldn't be there" (Health worker 07). Some FCHVs and health workers described being ostracised like a low caste person, which affected their ability to provide services: "We were looked upon and treated like untouchables" (FCHV). This social exclusion also extended to family members of health workers: "People in the community gave us a hard time. While purchasing vegetables in the market they said that we have come here to spread COVID. It was hard for us to go out in public and we had to survive on dried vegetables. We could not walk in the road. Our family members were also taunted while walking on the road" (Health worker 11). FCHVs were unable to move freely around the community, and this hindered their ability to identify newly pregnant women: "Nobody would allow anyone to visit their home. It was difficult to figure out who was pregnant in the community" (FCHV).

**Lack of access to mobile phones.** Some health workers tried to contact pregnant women over the phone to encourage them to visit the health institution. This was problematic, as few pregnant women had access to phones and their husbands often received the call: "It was not possible for us to call everyone. Pregnant women gave phone numbers to us, but it was difficult to reach them because sometimes their husband or son received the call. Some gave the contact number of their neighbour" (Health worker 06). Pre-pandemic, women without access to phones could be reached through outreach or home visits, but this was not possible during the pandemic. Virtual counselling was not possible because of limited internet and smart phone access.

## Discussion

ANC was severely disrupted during the pandemic. Health workers had to find ways to provide ANC with insufficient PPE and guidance whilst facing extreme levels of stigmatisation. Pregnant women were fearful or unable to visit health institutions during the pandemic because of COVID-19 control measures. Pre-pandemic and during the pandemic health workers tried to contact pregnant and postpartum women and families over the phone, but this was challenging because of limited access to phones, and required at least one ANC visit to collect their phone number. The pandemic prevented new pregnancies from being registered, and therefore the possibilities to provide services over the phone for these pregnancies were limited.

### Maternal health services in Nepal during the pandemic

There have been few studies in Nepal about how maternal health services were provided during the pandemic. One study of community members and a few health workers in province 2 also reported that pregnant women had difficulty accessing services because of COVID-19 control measures and closure of health facilities [39]. A study of 10 nurses in a hospital in urban Biratnagar in Nepal also found that fear of infection and passing COVID-19 to their family members affected care, and there were difficulties of staffing with redeployment and isolation of those infected [40]. Stigma was also discussed as affecting the mental health of health workers who faced undue pressure from family and friends to take leave or leave their

jobs during COVID-19 [41–43]. Khanal et al. reported that nurses had significantly higher odds of being fearful of COVID-19 than other health workers in the early phases of the pandemic [41].

## Quality of care and trust

Our study has shown that health workers were uncomfortable providing sub-optimal services to pregnant women, and that women were also dissatisfied. The benefits of ANC are clear [44], and there may be a danger that the gains made in increasing uptake of ANC in Nepal could be reversed if community perceptions about the need for ANC or the quality of ANC change as a result of the pandemic. Perceptions about the quality of IFA and ANC are important factors which affect access to care in Kapilvastu [19], and there is a need to maintain or re-gain trust in health institutions post-pandemic [45]. Our study shows a need to intervene at both the demand and supply side to maintain access and provision of care during emergencies such as the COVID-19 pandemic.

## Community engagement

Health workers showed that they were willing to provide care if they could protect themselves and their clients. The sense of commitment, and duty to provide care to pregnant women has also been documented in other research with nurses and midwives in LMICs [46, 47]. Co-ordinated community engagement through community leaders, mother's groups, radio, loud speaker, mobile messaging and social media are recommended strategies to address stigma and enable health workers to provide care [48]. Public recognition of the role of health workers and timely response to their concerns should be prioritised in the future to reward and motivate them.

## Access to PPE

There is a need to ensure equal and adequate access to PPE. Whilst lack of access to PPE during the pandemic was a global issue [49, 50] which put health workers at risk [51, 52], procurement in Nepal was problematic pre-pandemic, and has been identified as a bottleneck to the provision of quality maternal and newborn care [53]. Pre-pandemic, researchers have noted an urgent need to strengthen procurement systems as the health system decentralises. It has been reported that local levels lack capacity in this area [54, 55]. During the pandemic, research found that the lack of coordination between local, provincial, and federal levels of government weakened pandemic preparedness and affected procurement [56]. Our study also shows that health facilities had difficulties obtaining resources essential for comprehensive ANC, and robust procurement and distribution systems are necessary to deal with future emergencies.

## Context appropriate guidance

Health workers felt unprepared to deal with the pandemic and felt largely unsupported. Guidance on how to provide ANC was often difficult to follow in the community setting, where there was diverse and often inadequate physical infrastructure pre-pandemic, unsuited to guidance given during the pandemic. Explicit guidance, based on the experience of health workers working in a diversity of health institutions, is necessary for future resilience planning. Specifically on maintaining distancing, ventilation, infection prevention and dispensing to minimise risk to health workers and pregnant women. This guidance should also be communicated clearly across the three tiers of government and to health workers.

## Opportunities for mHealth

Before and during the pandemic, health workers were registering the mobile numbers of pregnant women. Telephone consultation was only possible with some pregnant women, and those who are most marginalised and those who would benefit most from ANC were unlikely to have access to a phone. Difficulties in locating newly pregnant women also limited the benefits of telephone consultation during the pandemic. mHealth is often promoted as a way to improve access to maternal and newborn care in LMICs [57], but we found that gender and socio-economic inequalities limited the usefulness of mHealth during the pandemic. Community engagement to ensure that families exchange phone numbers with their FCHV could help detect new pregnancies in situations when it is difficult to visit households.

## Limitations

We collected data in one district of Nepal, with a limited number of health workers which has limited our ability to explore similarities and differences in responses by gender and seniority. Responses may have been biased, with health workers and FCHVs hesitant to share sub-optimal practices. Data show that health workers felt comfortable to speak candidly, and we do not think this was a major limitation of our study.

## Conclusion

The pandemic affected access to and quality of ANC which meant that many women were not exposed to interventions which have improved maternal and newborn survival in LMICs. To address supply side issues, there is a need to establish and maintain robust procurement systems and improve coordination and communication between the three tiers of government and health workers. Health worker participation in the development of guidance reflecting the diversity of care settings is also recommended. mHealth interventions during pregnancy need to ensure access among the most marginalised and newly pregnant women who may not have access to a phone. To address demand side issues, community engagement is important to ensure women and families are aware of the need to access ANC and IFA, and to prevent stigmatisation of health workers. Addressing both demand and supply side issues will help to minimise disruptions to ANC in future health emergencies.

## Supporting information

**S1 Questionnaire.**
(DOCX)

## Acknowledgments

We would like to thank all the participants in the study. We would like thank field researchers Mr Basudev Bhattarai, Ms Anjali Basnet, Ms Bidhya Chaudhary.

## Author Contributions

**Conceptualization:** Sanju Bhattarai, Abriti Arjyal, Joanna Morrison.

**Data curation:** Bibhu Thapaliya, Samata Kumari Yadav, Santosh Giri, Suprich Sapkota.

**Formal analysis:** Bibhu Thapaliya, Samata Kumari Yadav, Santosh Giri, Suprich Sapkota, Joanna Morrison.

**Funding acquisition:** Helen Harris-Fry, Naomi Saville, Sara Hillman, Sushil Baral, Joanna Morrison.

**Investigation:** Bibhu Thapaliya, Samata Kumari Yadav.

**Methodology:** Sanju Bhattarai, Abriti Arjyal, Joanna Morrison.

**Project administration:** Sanju Bhattarai, Abriti Arjyal, Sushil Baral.

**Supervision:** Sara Hillman, Sushil Baral.

**Validation:** Bibhu Thapaliya, Samata Kumari Yadav, Sanju Bhattarai, Joanna Morrison.

**Visualization:** Bibhu Thapaliya, Samata Kumari Yadav, Joanna Morrison.

**Writing – original draft:** Bibhu Thapaliya, Joanna Morrison.

**Writing – review & editing:** Bibhu Thapaliya, Samata Kumari Yadav, Sanju Bhattarai, Santosh Giri, Suprich Sapkota, Abriti Arjyal, Helen Harris-Fry, Naomi Saville, Sara Hillman, Sushil Baral, Joanna Morrison.

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
