## [Decision Letter · Decision Letter 0]

17 Feb 2023

PONE-D-22-31111Health worker perspectives on access to antenatal care in rural plains Nepal during the pandemicPLOS ONE

Dear Dr. Morrison,

Thank you for submitting your manuscript to PLOS ONE. After careful consideration, we feel that it has merit but does not fully meet PLOS ONE’s publication criteria as it currently stands. Therefore, we invite you to submit a revised version of the manuscript that addresses the points raised during the review process.

Please see the reports from two reviewers below. Both reviewers have requested additional detail or clarification, particularly in the Methods section.

We look forward to receiving your revised manuscript.

Kind regards,

Hanna Landenmark

Staff Editor

PLOS ONE

Journal Requirements:

Reviewers' comments:

Reviewer's Responses to Questions

**Comments to the Author**

1. Is the manuscript technically sound, and do the data support the conclusions?

Reviewer #1: Yes

Reviewer #2: Yes

2. Has the statistical analysis been performed appropriately and rigorously? 

Reviewer #1: Yes

Reviewer #2: I Don't Know

3. Have the authors made all data underlying the findings in their manuscript fully available?

Reviewer #1: No

Reviewer #2: No

4. Is the manuscript presented in an intelligible fashion and written in standard English?

Reviewer #1: Yes

Reviewer #2: Yes

5. Review Comments to the Author

Reviewer #1: Overall, well written and well executed project.

I have these minor comments

As lines 61 - 128 are providing context on the situation in Nepal, I think it's best suited to be placed under "Setting" from line 131 as a "General Setting" subheading and then a subsequent "Specific Setting" subheading provided for the current information provided from line 132 - 161.

For Clarity; lines 200 to 202 can have "Ethical consideration" as a subheading

At the result section, the qualitative results and quantitative results are not immediately distinct. Including a table especially for the quantitative aspect of the study could help bring out clearly the outcomes of this aspect of your mixed method study.

Line 403 should be checked for clarity of thought and if needed broken into two sentences. Does "during, ..."relate to access to the phone or to ANC?

Reviewer #2: REVIEWERS COMMENTS

The study is objective is good. There are few comments however to be answered.

ABSTRACT

1.LINE 14-16: Not sure what you mean by health workers and community health volunteers. Aren’t community health volunteers part of health workers?

MATERIAL AND METHODS

SETTINGS

2.Are there health facilities in the district that provide above primary health care? Were they included in the study?

QUALITATIVE SAMPLING

3a. Line 164: How many health posts were in the district? Why did you opt for the 8 you used in the study?

3b. How were the interviews conducted? Direct or over the phone

QUANTITATIVE SAMPLING

4a.Line 188: How did you sample the 52 FCHVs for the study? What do you mean by 52 were available for study? Were the others under quarantine or they simply refused participation?

4b. Did you conduct direct interviews?

4c. line 200: Why verbal consent by some participants? Were they not accessible?

RESULTS

5. Line 325: I get the impression that tele medicine existed prior to the COVID-19 pandemic, didn’t you have the problem of lack of mobile phone access prevailing pre-pandemic? If it existed before the pandemic, how severe was it during the pandemic era?

LIMITATION

6. Line 413: the sentence should be looked at again. “Has” for ‘his’?

DISCUSSION

7. Line 387: I wish your quantitative data had covered other areas beyond IFA such as how many health posts lacked PPEs, how many PPEs were requested by each facility studied and how many were supplied. Without these facts, the statement on line 388 becomes questionable.

REFERENCES

8. Line 452: Not sure the journal which published the case study

9. Line 474: 2078? An error in the year?

10. Line 490: can you take a look again at the reference, 2064?

11. Line 585: the date for the publication

6. PLOS authors have the option to publish the peer review history of their article (what does this mean?). If published, this will include your full peer review and any attached files.

Reviewer #1: No

Reviewer #2: No

---

## [Author Response · Author response to Decision Letter 0]

11 Mar 2023

Reviewer #1: Overall, well written and well executed project.

Response: Thank you for appreciating our paper and project.

I have these minor comments

As lines 61 - 128 are providing context on the situation in Nepal, I think it's best suited to be placed under "Setting" from line 131 as a "General Setting" subheading and then a subsequent "Specific Setting" subheading provided for the current information provided from line 132 - 161.

Response: We do not discuss the specific setting of the study in the introduction section, and headings mention the country context (Nepal). We have added to methods section heading we have added ‘study setting’ to clarify that this paragraph describes the specific context of the study.

For Clarity; lines 200 to 202 can have "Ethical consideration" as a subheading.

Response: We have added the subheading “Ethical consideration” to the mentioned lines as per your suggestion. 

At the result section, the qualitative results and quantitative results are not immediately distinct. Including a table especially for the quantitative aspect of the study could help bring out clearly the outcomes of this aspect of your mixed method study.

Response: We have disaggregated our quantitative and qualitative sample in the methods section. Our quantitative data are not extensive, and therefore we feel they do not warrant a separate section.

Line 403 should be checked for clarity of thought and if needed broken into two sentences. Does "during, ..."relate to access to the phone or to ANC?

Response: Thank you for your suggestion. We have changed the line 403 to make it clearer.

Reviewer #2: REVIEWERS COMMENTS

The study is objective is good. There are few comments however to be answered.

ABSTRACT

1.LINE 14-16: Not sure what you mean by health workers and community health volunteers. Aren’t community health volunteers part of health workers? 

Response: Thank you for your comment. Female Community Health Volunteers have no formal qualifications, and their role is mainly health promotion as described on page 4. They are a distinct cadre separate from health workers.

MATERIAL AND METHODS

SETTINGS

2.Are there health facilities in the district that provide above primary health care? Were they included in the study?

Response: There were hospitals providing higher levels of health care in the district headquarters. These were not included in this study because we chose to focus our research on primary health care only. 

QUALITATIVE SAMPLING

3a. Line 164: How many health posts were in the district? Why did you opt for the 8 you used in the study?

Response: We sampled the eight health posts because we were already in contact with health workers from those health posts for another study (which has now been published and is referenced in the paper Saville et al. 2023). We have added the words ‘conveniently sampled’ to clarify that our sampling of health posts was convenient. 

3b. How were the interviews conducted? Direct or over the phone

Response: The interviews were conducted directly, and we have added this detail to the paper on age 9.

QUANTITATIVE SAMPLING

4a.Line 188: How did you sample the 52 FCHVs for the study? What do you mean by 52 were available for study? Were the others under quarantine or they simply refused participation?

Response: There are a set number of FCHVs working under each health post. We sampled half the FCHVs working under the eight health posts. We have deleted the sentence about their availability as we agree with reviewers that it is confusing.

4b. Did you conduct direct interviews?

Response: Yes, we conducted direct interviews, and we have added this to the text.

4c. line 200: Why verbal consent by some participants? Were they not accessible?

Response: Some participants had low literacy and therefore we took verbal consent to participate from those participants. We have added to the text to clarify this.

RESULTS

5. Line 325: I get the impression that tele medicine existed prior to the COVID-19 pandemic, didn’t you have the problem of lack of mobile phone access prevailing pre-pandemic? If it existed before the pandemic, how severe was it during the pandemic era?

Response: FCHVs and health workers were meant to interact with pregnant women over the phone pre-pandemic. This was problematic before and during the pandemic because not all women had access to phones. Pre-pandemic FCHVs and health workers could interact face to face with women who didn’t have access to phones, but with lockdown preventing movement this was challenging. We have added to this paragraph on page 17 to clarify why lack of access was particularly troubling during the pandemic.

LIMITATION

6. Line 413: the sentence should be looked at again. “Has” for ‘his’?

Response: We have corrected this in our manuscript. 

DISCUSSION

7. Line 387: I wish your quantitative data had covered other areas beyond IFA such as how many health posts lacked PPEs, how many PPEs were requested by each facility studied and how many were supplied. Without these facts, the statement on line 388 becomes questionable.

Response: We have removed the part about PPE in line 388.

REFERENCES

8. Line 452: Not sure the journal which published the case study

Response: Thank you for your comment. We have corrected the references for this case study. 

9. Line 474: 2078? An error in the year? 

10. Line 490: can you take a look again at the reference, 2064?

Response: The documents are published under the Nepali calendar, and the reference years are correct. If we changed the reference years, readers would not be able to find the references.

11. Line 585: the date for the publication

Response: We have added the date of publication.

---

## [Editor Report · Decision Letter 1]

10 Apr 2023

Health worker perspectives on access to antenatal care in rural plains Nepal during the COVID-19 pandemic

PONE-D-22-31111R1

Dear Dr. Morrison,

We’re pleased to inform you that your manuscript has been judged scientifically suitable for publication and will be formally accepted for publication once it meets all outstanding technical requirements.

Kind regards,

Seth Amponsah-Tabi

Guest Editor

PLOS ONE

Additional Editor Comments (optional):

Dear, the revised session is much better. I deem your responses to comments as appropriate.
---

## [Editor Report · Acceptance letter]

14 Apr 2023

PONE-D-22-31111R1 

Health worker perspectives on access to antenatal care in rural plains Nepal during the COVID-19 pandemic 

Dear Dr. Morrison:

I'm pleased to inform you that your manuscript has been deemed suitable for publication in PLOS ONE. Congratulations! Your manuscript is now with our production department. 

Kind regards, 

on behalf of

Dr. Seth Amponsah-Tabi 

Guest Editor

PLOS ONE